# Primary and Metastatic Brain Tumours Assessed with the Brain and Torso [^18^F]FDG PET/CT Study Protocol—10 Years of Single-Institutional Experiences

**DOI:** 10.3390/ph14080722

**Published:** 2021-07-26

**Authors:** Agata Pietrzak, Andrzej Marszałek, Tomasz Piotrowski, Adrianna Medak, Katarzyna Pietrasz, Julia Wojtowicz, Hubert Szweda, Krzysztof Matuszewski, Witold Cholewiński

**Affiliations:** 1Electroradiology Department, Poznan University of Medical Sciences, 61-866 Poznan, Poland; tomasz.piotrowski@me.com (T.P.); adrianna.medak@op.pl (A.M.); witoldc@onet.pl (W.C.); 2Nuclear Medicine Department, Greater Poland Cancer Centre, 61-866 Poznan, Poland; katarzyna.pietrasz@wco.pl; 3Oncologic Pathology and Prophylaxis Department, Poznan University of Medical Sciences and Greater Poland Cancer Center, 61-866 Poznan, Poland; amars@ump.edu.pl; 4Medical Physics Department, Greater Poland Cancer Centre, 61-866 Poznan, Poland; hubert.szweda@wco.pl (H.S.); krzysztof.matuszewski@wco.pl (K.M.); 5Medical Faculty, Lodz Medical University, 90-647 Lodz, Poland; julia.wojtowicz@stud.umed.lodz.pl

**Keywords:** brain tumour, ^18^F-fluorodeoxyglucose, oncology, positron emission tomography

## Abstract

According to the international societies’ recommendations, the 2-deoxy-2-[^18^F]fluoro-D-glucose positron emission tomography/computed tomography ([^18^F]FDG PET/CT) technique should not be used as the method of choice in brain tumour diagnosis. Therefore, the brain region can be omitted during standard [^18^F]FDG PET/CT scanning. We performed comprehensive literature research and analysed results from 14,222 brain and torso [^18^F]FDG PET/CT studies collected in 2010–2020. We found 131 clinically silent primary and metastatic brain tumours and 24 benign lesions. We concluded that the brain and torso [^18^F]FDG PET/CT study provides valuable data that may support therapeutic management by detecting clinically silent primary and metastatic brain tumours.

## 1. Introduction

Brain tumours are relatively rare malignancies, approximating 2% of all oncologic diseases. Currently, the incidence of the primary and metastatic brain tumours seems to be especially increasing in highly developed countries, and the oligometastatic brain disease seems to be especially concerning [1,2,3]. Malignant brain tumours are often deadly and involve several neurological and locomotory health ailments. Thus, both primary and metastatic brain tumour patients undergo treatment. Most often, therapy involves chemotherapy or chemoradiation rather than surgery. Histologic examination is not always available, due to the tumour location and the high postsurgical mortality risk among brain tumour patients. Therefore, prompt and complex diagnosis seems essential [1,3,4,5,6].

Central nervous system (CNS) disease diagnostic management involves mainly magnetic resonance imaging (MRI). However, 2-deoxy-2-[^18^F]fluoro-D-glucose positron emission tomography/computed tomography ([^18^F]FDG PET/CT) study remains one of the most used imaging techniques in oncology. According to the international societies’ recommendations, the non-tumour-specific properties of the radiotracer [^18^F]FDG limit the specificity of the method in brain tumour detection and the CNS benign versus (vs.) malignant lesion differentiation. Therefore, the brain region is often omitted in the standard [^18^F]FDG PET/CT scanning protocol [7,8,9].

In this study, we analysed the recommended [^18^F]FDG PET/CT study performance principles, published by the widely respected organisations, International Atomic Energy Agency (IAEA) [10], the European Association of Nuclear Medicine (EANM) [11,12], and the Society of Nuclear Medicine and Molecular Imaging (SNMMI) [13,14], as well as additional sources of recommendations [15,16,17,18]. We discussed the [^18^F]FDG PET/CT method applications and limitations in the diagnosis of clinically silent brain tumours, considering international guidelines and our experiences in performing brain and torso [^18^F]FDG PET/CT, to discuss whether the brain region should be included in the standardly performed acquisition protocol.

## 2. Materials and Methods

### 2.1. Bioethics

This study was designed per receipt of the patients’ written informed consent and approved by the Local Bioethical Committee (Poznan University of Medical Sciences Bioethical Committee, chair: Paweł Chęciński, date of approval: 30 January 2021) as the retrospective analysis based on standardly performed procedures, conducted in 2010–2020. All data have been strictly anonymized.

### 2.2. Literature Collection

In this study, we analysed the IAEA [10], EANM [11,12], and the SNMMI [13,14] original and updated guidelines, describing the utilities of the [^18^F]FDG PET/CT study in brain region imaging. We used the additional materials referring to the role of the [^18^F]FDG PET/CT examination in brain tumour diagnosis: the World Health Organisation (WHO) reports [1,19], the *National Tumour Brain Society* publication [3], and the EANM educational materials [15,16]. We compared the above-mentioned sources of recommendations and we discussed our experiences in performing the brain and torso [^18^F]FDG PET/CT study, considering the ability of the method to detect clinically silent brain tumours. The study protocol mentioned in the manuscript can be recognized as the whole-body or the brain and torso acquisition [11]. The used protocol can be described as the scanning ranging from the skull apex (vertex, top of the head) to mid-thigh. The main difference between the most used scanning protocols and the brain and torso imaging is the inclusion of the brain region, which is very often omitted in daily [^18^F]FDG PET/CT study performance. Figure 1 shows the flow diagram of the literature search method.

### 2.3. Original Database

We evaluated 14,222 consecutive [^18^F]FDG PET/CT datasets obtained in 2010–2020, considering the occurrence of clinically silent brain lesions. We followed the following inclusion criteria: single-phase brain and torso [^18^F]FDG PET/CT imaging performed, no brain foci mentioned in the patients’ medical records, at least 1 year follow-up (range: 1–7 years), treatment applied in our institution. We analysed medical records and other imaging studies conducted in those patients. We have excluded from the final analysis those patients in whom the suspicious brain findings were reported previously in the medical records. In some examined subjects, the histopathologic examination confirming brain malignancy was not available due to the tumour’s location and a high postsurgical mortality risk. In those conditions, we used the data obtained with repeated brain and torso [^18^F]FDG PET/CT study and MRI. We found suspicious, clinically silent brain findings in 155 patients. Figure 2 shows the original database collection scheme.

Based on patients’ medical records, we divided the population of 155 examined subjects into the following groups, considering the detected brain lesion type: benign lesions, primary brain tumours, metastatic foci. We presented and discussed the epidemiological characteristics of the above-mentioned groups in the context of worldwide brain lesion incidence.

### 2.4. Equipment, Software, Measurements

We used the Philips Gemini TF16 hybrid scanner (Philips, Cleveland, OH, USA) to perform the brain and torso [^18^F]FDG PET/CT [11] study, ranging from the skull apex to mid-thigh [11,20]. Before the scanning, we asked patients to keep warm and hydrate, starting at 1 hour (h) prior to the injection of the radiotracer. We performed the study at approximately 60 min post-injection (p.i.) of the radiopharmaceutical [^18^F]FDG in activity up to 3.7 MBq per kilogram (kg) of the patient’s body weight (BW), after plasma glucose level evaluation (acceptable level: <120 mg/dL) [20]. PET/CT study included body low-dose CT using the following technical conditions: 100–200 milliampere-seconds (mAs), 120 kilovoltage peak (kVp), pitch of 0.8, X-ray tube rotation of 0.5 s. We performed PET acquisition using the 90 s per section scanning. Study duration did not exceed 25–35 min [20].

We obtained the PET-dedicated metabolic parameter of the maximal standardised uptake value (SUVmax) to assess brain lesions’ glucose metabolism activity. SUV describes a utilization (radiotracer tissue concentration) in a tumour based on a distribution volume as follows [20]:SUV = Activity_voi_ [kBq/mL]/(Activity_administered_ [MBq]/BW [kg])

We used the semiautomatic contouring method to evaluate brain lesions, using the following applications: Fusion Viewer (Philips, Cleveland, OH, USA) and MiM 7.0 (MiM Software Inc., Cleveland, OH, USA) available upon the single-institutional license.

### 2.5. Statistical Analyses

We evaluated the SUVmax value within the analysed groups of lesions and compared those groups considering the mean SUVmax value levels. Before we performed the necessary statistical analyses, we used the Shapiro–Wilk normality test and divided datasets into independent groups of lesions. In this study, we followed the statistical significance level of α = 0.05. We interpreted the statistical tests’ results, considering *p*-value. We used the Mann–Whitney’s U-test to compare the SUVmax value levels obtained within brain lesions. During the statistical analyses and forming of the conclusion, we followed the assumptions of null and alternative hypotheses (H_0_, H_a_, respectively): H_0_ suggested that the true variables’ distribution was normal or the evaluated differences between the calculations were statistically insignificant (*p* > 0.05); H_a_—true distribution significantly differed from Gaussian, or we observed insignificant differences (*p* < 0.05) [21].

We used the *STATISTICA 13.3* software (StatSoft; TIBCO Software, Palo Alto, CA, USA, available upon the individual license) to perform statistical analyses.

## 3. Results

In this study, we performed a comprehensive analysis of the international societies’ recommendations considering the utilities of the [^18^F]FDG PET/CT study in brain lesion evaluation. We focused the attention on the role of [^18^F]FDG in obtaining both benign and malignant brain lesions. We included in the analysis all detected benign, primary, and metastatic brain findings. We briefly mentioned our database’s epidemiological characteristics and described the obtained brain foci and the most recommended radiopharmaceuticals for brain tumour evaluation. We compared the international guidelines with our experiences in performing brain and torso [^18^F]FDG PET/CT [11] examination to conclude whether the [^18^F]FDG PET/CT study protocol should include brain imaging.

### 3.1. The Role of the [^18^F]FDG PET/CT Study in CNS Evaluation—Recommendations

Commonly used [^18^F]FDG PET/CT acquisition protocol ranges from the skull-base to mid-thigh [10,11,12,13,14,15,16,17,18]. Boellard et al. described the brain and torso [^18^F]FDG PET/CT [11] study protocol as a modification of the standard examination. The authors [10,11,12,13,14,15,16,17,18] discussed the potential utilities of [^18^F]FDG PET/CT imaging in brain disease diagnosis, including non-invasive staging for therapy planning. The main limitation of the [^18^F]FDG PET/CT study is the non-tumour-specific properties of the radiotracer [^18^F]FDG. According to the IAEA, EANM, and SNMMI recommendations [10,11,12,13,14,15,16], a high glucose uptake within the grey matter significantly decreases the specificity of the study in detecting small lesions within the CNS and distinguishing benign (i.e., inflammation) and malignant tumours. Nevertheless, IAEA [10] mentions the delayed PET/CT imaging at 4–6 h p.i. of [^18^F]FDG as improving the specificity of the method by increasing the tumour-to-background ratio. However, we did not mention delayed protocol in this study (Table 1).

According to the authors [18,22,23,24,25,26], the PET-dedicated radiopharmaceuticals of choice in brain lesion diagnosis are [^11^C]-methionine ([^11^C]MET), [^18^F]-fluoroethyltyrosine ([^18^F]FET), [^18^F]-3′-deoxy-3′-fluorothymidine ([^18^F]FLT), and [^18^F]-dihydroxyphenylalanine ([^18^F]DOPA). Jung et al. [18] recommend using [^11^C]MET, [^18^F]FLT, [^18^F]DOPA to improve the sensitivity and specificity of the PET/CT method in low-grade glioma imaging. Moreover, [^18^F]DOPA has been recognised as particularly useful in remnant brain tumour diagnosis in patients who underwent surgery or both tumour resection and radiotherapy [22,23,24,25]. Authors [18,22,23,24,25] mention a high [^18^F]FDG uptake within the grey matter, as well as the limited ability of the [^18^F]FDG PET/CT method in benign vs. malignant CNS lesion differential diagnosis. According to the literature [18,22,23,24,25], the most suitable radiopharmaceuticals for brain tumour diagnosis are cell proliferation markers.

### 3.2. Original Database—Brain and Torso [^18^F]FDG PET/CT Study

#### 3.2.1. Epidemiology

We examined 14,222 patients using the brain and torso [^18^F]FDG PET/CT. We found suspicious brain lesions in 155 patients. The group consisted of 96 women and 59 men; mean age ± standard deviation (S.D.) was 60 ± 12 years old (y.o.), age range: 26–84 y.o. Among women, the mean age ± S.D. was 60 ± 12 y.o., range: 26–84 y.o.; men: 61 ± 13 y.o., range: 26–84 y.o.

The benign lesion group consisted of 15 women (mean age ± S.D.: 63 ± 13 y.o., range: 35–84 y.o.) and 9 men (mean age ± S.D.: 60 ± 13 y.o., range: 40–83 y.o.). The primary brain tumour group consisted of 16 women (mean age ± S.D.: 64 ± 11 y.o., range: 46–81 y.o.) and 12 men (mean age ± S.D.: 63 ± 17 y.o., range: 26–84 y.o.). The metastatic foci group consisted of 65 women (mean age ± S.D.: 58 ± 12 y.o., range: 30–78 y.o.) ans 38 men (mean age ± S.D.: 61 ± 11 y.o., range: 39–81 y.o.). In our study, we observed the highest brain finding incidence among women over the age of 60 y.o.

#### 3.2.2. Benign Lesions

We evaluated 24 benign lesions in the brain region: 22 arachnoid cysts, 1 adenoma, and 1 unspecified vascular malformation. In one patient, the arachnoid cyst was one of a few brain foci (renal cancer metastasis). Arachnoid cysts determined 87.5% of all detected benign lesions and occurred in 13.5% of the examined 155 patients.

We detected benign brain lesions in patients diagnosed with breast cancer (five patients), as well as colorectal (four), cervical (three), ovarian (two), adrenal (one), adrenal (one) and renal (one), prostate (one), testicular (one), liver (one; hepatocellular cancer; HCC), and lung (one) cancer. In two subjects, the indications to perform the brain and torso [^18^F]FDG PET/CT scanning included malignant melanoma staging.

In this study, we observed a relatively low glucose metabolism activity within the benign brain lesions. The mean SUVmax ± S.D. was 1.0 ± 0.2, the median (M_e_) was 1.0, range: 1.0–2.0 (CI_95 =_ (1.1; 1.3)). The variables were normally distributed (*p* = 0.8). Thus, the benign lesions group was homogenous, considering the metabolic activity of the observed lesions. Figure 3 shows the arachnoid cyst.

#### 3.2.3. Primary Brain Tumours

We incidentally detected 28 primary brain tumours (18% of primary tumours and 21% of all malignant lesions). In this group, we found 10 gliomas, 8 pituitary gland tumours, 4 malignant meningiomas, 4 cerebelli primaries, 1 brain lymphoma, and 1 base of the skull tumour. In some of the examined patients, we observed multiple lesions (up to six foci). Gliomas determined nearly 36% of all brain primaries.

The primary brain lesions occurred in patients [^18^F]FDG PET/CT diagnosed with cancer of unknown primary (CUP syndrome; 18 patients), as well as colorectal (three patients), breast (one), cervical (one), oesophagal (one), lung (one), prostate (one), and uterine (one) cancer. In one patient, the primary disease was malignant melanoma.

We observed widely ranged glucose metabolism activity levels within the primary brain tumour group due to the presence of both non-[^18^F]FDG-avid and low- and high-grade tumours in the datasets. The mean SUVmax ± S.D. was 9.2 ± 4.7, the M_e_ was 9.0, range: 1.2–25.0 (CI_95_ = (7.3; 11.0)). Figure 4 and Figure 5 show the primary brain tumours.

#### 3.2.4. Metastatic Foci

We detected 103 metastatic brain foci in 27 breast cancer patients (in one case, simultaneous breast and lung cancer), 20 lung cancer subjects (in one case, concurrent lung and colorectal cancer), malignant melanoma (25), colorectal cancer (nine), ovarian (four), renal (three), gastric (three), prostate (two), uterine (two), urinary bladder cancer (one), thyroid (two), and pancreas tumour (one), as well as Hodgkin’s (one) and non-Hodgkin’s lymphoma (three).

The oligometastatic brain disease (up to eight foci) occurred in nearly 70% of the metastatic foci database. Accordingly, with the patients’ medical records, in 45% of the group (46 among 103 cases), none other than brain lesions were detected. In approximately 10%, brain foci were one of the few observed lesions. In 35%, we found significant primary disease spread with multiple distant tumours with advanced lymph node involvement.

The metastatic foci group consisted of the most metabolically active malignant brain lesions. The mean SUVmax ± S.D. was 12.4 ± 5.6, the M_e_ was 12.0, and SUVmax range was 4.0–33.0 (CI_95_ = (11.3; 13.5)). The SUVmax distribution significantly differed from Gaussian, with *p* < 0.001. We observed the highest SUVmax levels within the lung cancer and the malignant melanoma metastases. In this study, we observed the highest glucose metabolism activity in the metastatic foci group. Figure 6 shows the metastatic brain lesion.

### 3.3. Summary

Performing brain and torso [^18^F]FDG PET/CT examination resulted in detecting clinically silent malignant brain lesions in 155 patients diagnosed with different oncological diseases. The most numerous group was the metastatic foci, characterised by the highest mean SUVmax levels. When compared with the primary brain lesions, the distant tumour’s SUVmax level was significantly higher, with *p* = 0.003 (Mann–Whitney’s U-test). However, the differences in sample size limit the reliability of this comparison and do not allow the performing of benign and malignant tumour differential diagnosis analysis. Table 2 shows the SUVmax measurements.

## 4. Discussion

According to IAEA (2013) [10], EANM (2009, 2015) [11,12], and SNNMI (2006, 2009) [13,14] recommendations, the brain region is not an obligatory element of the standard [^18^F]FDG PET/CT scanning protocol. The area of skull apex to mid-thigh seems to be a protocol applied to some groups of patients, i.e., those of a high risk of skull and brain metastasis. EANM recommendations indicate the potential usefulness of the [^18^F]FDG PET/CT examination in monitoring brain tumour recurrent disease and non-invasive grading purposes. The authors indicated comparable limitations of the method, such as a high physiologic glucose uptake within the CNS (EANM) and a low ability to detect small lesions (SNNMI). Moreover, according to the IAEA, a high grey matter glucose uptake limits the specificity of the method in distinguishing inflammation and malignant tumours (Table 1).

The most recommended PET-dedicated radiopharmaceuticals for brain tumour diagnosis are [^11^C]MET, [^18^F]FET, [^18^F]FLT, and [^18^F]DOPA [10,15,23,24,25,26], as well as [^18^F]MISO and [^18^F]FAZA [10]. Nevertheless, the authors [10,11,12,13,14,15,16,17,18] suggest that performing brain and torso [^18^F]FDG PET/CT scanning [11] can be useful in high-grade brain tumours due to increased glucose uptake within the lesion. It can also detect the non-[^18^F]FDG-avid tumours (observed as the cold spot surrounded by the metabolically active brain cortex).

This study aimed to present the possibility to detect clinically silent malignant brain lesions using brain and torso [^18^F]FDG PET/CT examination. While collecting the original database, we have excluded from the final analysis those patients in whom the suspicious brain findings were reported previously. In some examined subjects, the histopathologic examination confirming brain malignancy was not available due to the tumour’s location and high postsurgical mortality risk. To ensure the reliability of the conducted research, we decided to focus on those patients in whom performing only the brain and torso [^18^F]FDG PET/CT provided any data regarding brain region malignancy. We excluded also the oncological patients transferred to another institution for further management due to the risk of a lack of sufficient curation of the data used in this research.

According to the *World Cancer Report 2020* [19], the most commonly diagnosed brain tumours are benign lesions and malignant gliomas. In this study, we detected arachnoid cysts in 22 patients (87.5% of all benign lesions), and gliomas in eight subjects (36% of brain primaries). Currently, the most concerning worldwide seems to be the increasing incidence of oligometastatic brain disease (observed in nearly 70% of the metastatic foci group).

In this study, the brain and torso [^18^F]FDG PET/CT helped to detect 155 clinically silent brain lesions among 14,222 examined patients. The average SUVmax among benign, primary, and metastatic brain lesions were 1.0, 9.2, 12.4, respectively (Table 2). We observed significantly higher glucose uptake of the metastatic foci when compared with the primary tumours. However, the difference in sample sizes limited the possibility to evaluate the usefulness of the [^18^F]FDG PET/CT study in distinguishing primary and metastatic brain lesions.

Due to high mortality among brain tumour patients, malignant brain foci very often undergo treatment. In this study, we presented data obtained in heterogenous and numerous group of consecutive patients who underwent the [^18^F]FDG PET/CT imaging, aiming to establish the current stage of the primary disease (among which: recurrence) and to detect distant tumours. Discovering clinically silent brain tumours seems to be especially important in patients in whom no distant metastases nor local recurrence were present. Among the metastatic foci patients, we found isolated brain tumours in 45%, single lesions localised in different regions in 10%, and highly advanced whole-body oligometastasis in 55% of examined patients.

The radiopharmaceutical [^18^F]FDG remains the most accessible and, sometimes, the only available for the institution radiotracer used for oncological purposes. Omitting brain imaging from the [^18^F]FDG PET/CT protocol decreases the possibility to evaluate clinically silent brain lesions and, therefore, hinders the implementation of the appropriate treatment. It seems to be especially important in patients with no other lesions than brain malignant lesions observed.

## 5. Conclusions

Performing the brain and torso [^18^F]FDG PET/CT study can provide valuable data supporting therapeutic management by detecting clinically silent primary and metastatic brain tumours.

## Figures and Tables

**Figure 1 pharmaceuticals-14-00722-f001:**
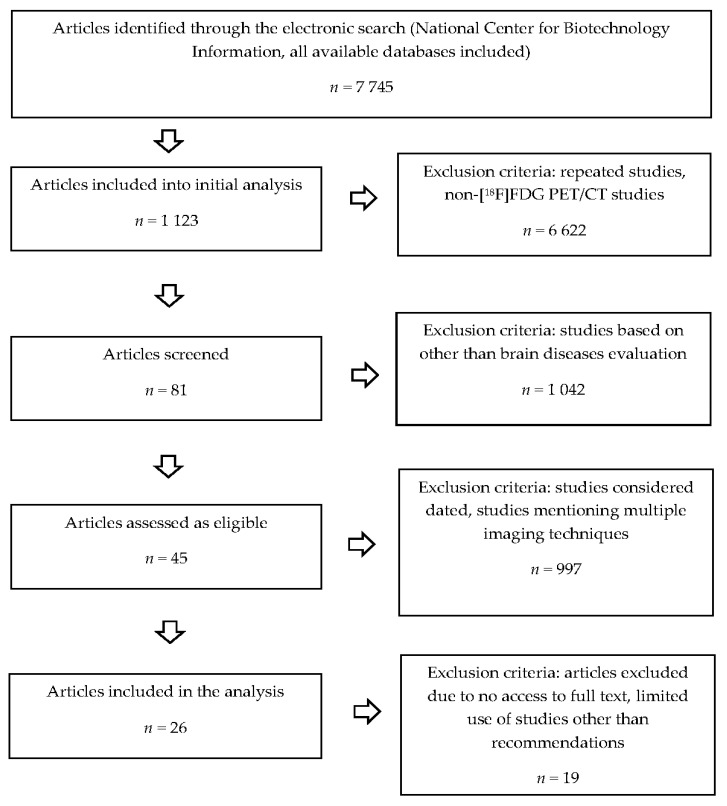
Flow diagram of literature search methods (source: original figure).

**Figure 2 pharmaceuticals-14-00722-f002:**
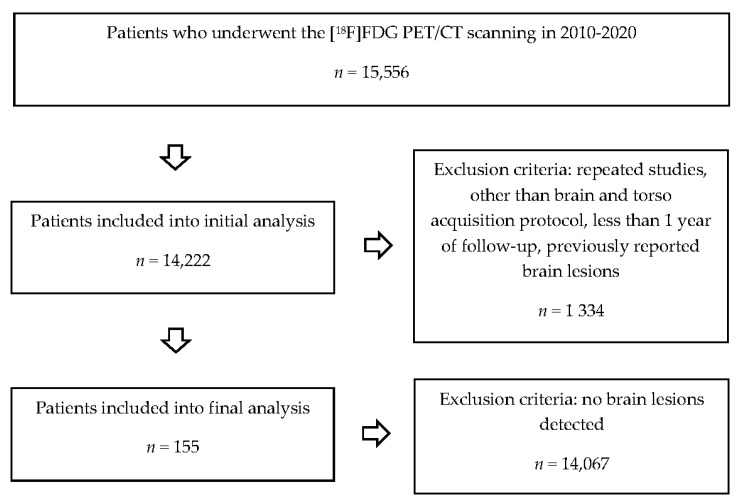
Flow diagram of database collection (source: original figure).

**Figure 3 pharmaceuticals-14-00722-f003:**
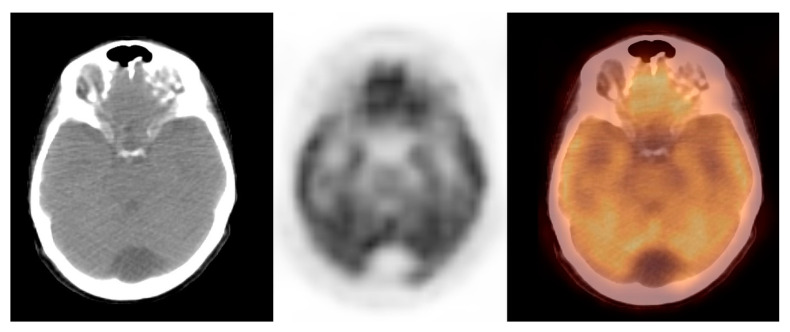
Incidental finding of a hypodense and photopaenic area in the posterior part of the brain in the middle cranial fossa, consistent with arachnoid cyst, in a patient with cervical cancer (source: original figure). Description: axial view of the brain and torso [^18^F]FDG PET/CT over the middle cranial fossa—low-dose CT (left-hand side image), PET (middle image), and PET/CT fusion (right-hand side image).

**Figure 4 pharmaceuticals-14-00722-f004:**
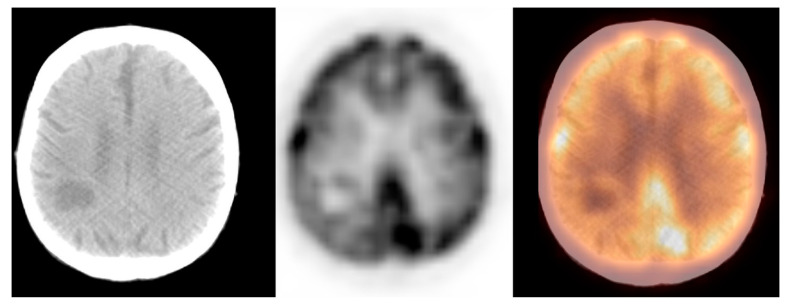
Incidental finding of a hypodense and photopaenic area in the right parietal lobe, consistent with primary brain tumour, in a patient with larynx cancer (source: original figure). Description: axial view of the brain and torso [^18^F]FDG PET/CT over the middle cranial fossa—low-dose CT (**left**-hand side image), PET (**middle** image), and PET/CT fusion (**right**-hand side image).

**Figure 5 pharmaceuticals-14-00722-f005:**
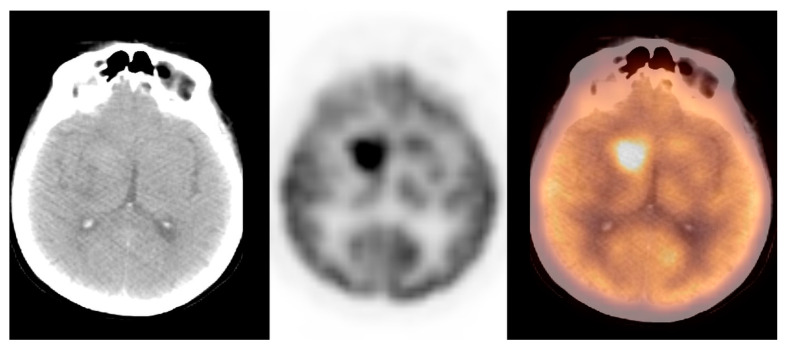
A hyperdense and hypermetabolic mass in the right thalamus region, consistent with primary brain tumour (glioblastoma), in a patient with breast cancer (source: original figure). Description: axial view of the brain and torso [^18^F]FDG PET/CT over the middle cranial fossa—low-dose CT (**left**-hand side image), PET (**middle** image), and PET/CT fusion (**right**-hand side image).

**Figure 6 pharmaceuticals-14-00722-f006:**
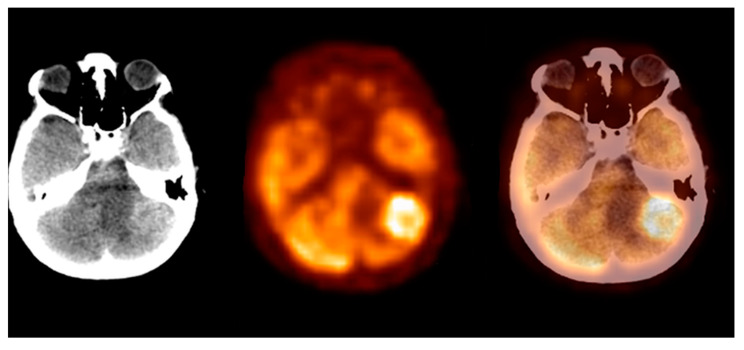
Incidental finding of a left lobe cerebellum abnormal mass, consistent with unknown and clinically silent lung cancer solitary metastatic lesion (source: original figure). Description: axial view of the brain and torso [^18^F]FDG PET/CT over the cerebellum—low-dose CT (**left**-hand side image), PET (**middle** image), and PET/CT fusion (**right**-hand side image).

**Table 1 pharmaceuticals-14-00722-t001:** [^18^F]FDG PET/CT in brain tumour diagnosis—international societies’ recommendations.

Source of Recommendations	Applications ^4^	Limitations ^5^
EANM ^1^ [11,12,15,16]	Non-invasive tumour grading	Low specificity in metastatic brain tumour evaluation
SNMMI ^2^ [13,14]	Benign and malignant brain tumour detection	Limited ability to assess small brain tumours
IAEA ^3^ [10]	Primary and metastatic brain tumour detection	Low specificity in distinguishing benign and malignant brain lesions

^1^ European Association of Nuclear Medicine (Europe). ^2^ Society of Nuclear Medicine and Molecular Imaging (United States of America). ^3^ International Atomic Energy Agency. ^4,5^ [^18^F]FDG PET/CT study applications and limitations in brain imaging.

**Table 2 pharmaceuticals-14-00722-t002:** SUVmax: benign lesions, primary brain tumours, metastatic foci (source: original data).

Group/Parameter	SUVmax ^1^ ± S.D. ^2^	SUVmax Median	SUVmax Range	CI_95_ ^3^
Benign lesions	1.0 ± 0.2	1.0	1.0–2.0	[1.1; 1.3]
Primary brain tumours	9.2 ± 4.7	9.0	1.2–25.0	[7.3; 11.0]
Metastatic foci	12.4 ± 5.6	12.0	4.0–33.0	[11.3; 13.5]

^1^ SUVmax—maximal standardised uptake value. ^2^ S.D.—standard deviation. ^3^ CI_95_—95% confidence interval (valid for at least 95% of studied population considering the SUVmax mean).

## Data Availability

Data sharing not applicable.

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
