# Peer review of "Primary and Metastatic Brain Tumours Assessed with the Brain and Torso [18F]FDG PET/CT Study Protocol—10 Years of Single-Institutional Experiences"

_pharmaceuticals, 2021, doi:10.3390/ph14080722_

Round 1

Reviewer 1 Report

Dear authors,

thank you very much for this request and opportunity to review an article for your journal.

I think that this paper is very interesting and the topic is surely original. The idea to analyze the main international guidelines and compered these guidelines with the clinical experience derived from many papers and our experience is original and interesting.

The type of acquisition (torso or whole body) in the clinical practice may be a dilemma for the nuclear medicine physicians and different point of view are present in literature.

However the methods are not clear, in particular the inclusion criteria need to be clarify both for articles and patients. Without these findings it is not so easy to correctly evaluate the results of this review. The discussion is very poor without specific explanation of the main results.

The reason of the inclusion of the literature articles is not clear.

Moreover, the conclusion are not supported by your data (0.01% of brain lesion detected with brain+dorso scan doesn't seem a great result).

METHODS

- The inclusion criteria of the articles are lacking. Which studies do you select? Studies with clinical suspect of brain lesions? Studies with "whole body" scan without any suspect? mixed? Please add some paragraphs with Search strategy, Study selection and Data abstraction.

Besides, I don't understand why do you include external articles if nothing about them is described in the manuscript?

- The inclusion criteria of your patients are lacking. What do you mean with "similar protocol"? Why did you performed torso and brain scan without clinical indications as you described?

- A table with the resume of the main technical features (type of scanner, uptake time, CT features, activities injected, and so on) of 24 article included is lacking. Besides, you should cite the articles that you used.

- "We chose the most often mentioned in the medical 113 literature statistical tests to obtain results and to provide conclusions" is too vague. Please clarify. Maybe, you should move paragraph 3.1 in the methods section.

- I suggest to add a table to resume the Results section to help the readers.

- I suggest to explain better in the discussion the findings of this work and less the general characteristics of brain tumors....

- Nothing about the relationship between size of brain lesion and PET/CT is reported. This is a crucial point. We know that the resolution power of PET scanner is about 5 mm. Moreover, we have the partial volume effect. So the detection of brain lesions and the SUV values are influenced intrinsically by these two variables. write something about this point in the discussion section.

- The reference 24 used as reference for suggesting SUV of 2.5 is not shareable. I suggest to delete this part. There is a strong debate in literature about the potential best cutoff of SUV and SUV is a factor affected by many parameters (body weight, technical features, extravasation, blood glucose level). I believe excessive this point.

- Why do you write that your data support a change in "therapeutic management "? in you paper you didn't analysed the change in treatment or outcome in the patients.

- You founded 0.01% of brain lesion among 14222 patients analysed. With these results underline the usefulness of brain scan is not shareable.....

REFERENCES

- You correctly spoke about not-FDG tracers for the study of brain, but I believe that you could add some more references about the usefulness of FDG in some brain cancers.

Moreover, a recent review about different tracers is available and to cite: doi: 10.3390/ijms20194669.

Reviewer 2 Report

This 10-years analysis is valuable to the field.

The major concerns of this analysis are

  1. It is still not clear when FDG-PET brain scan should be indicated or to be included along with the torso scan, for metastases?
  2. Primary brain tumors can also be non-FDG-avid. Without another tracer such as [C-11]methionine or [F-18]DOPA, how could this (cold spot) be certain to the primary brain tumor;
  3. The biggest concern is that among 14,222 patients in 10 years, how many patients with brain lesions are missed by FDG-PET given all the guidelines even though 155 were detected.

Minor concerns are

4. delayed scan was mentioned to differentiate inflammation, but no patient data/images were followed;

5. many "silent" primary brain tumors were detected from patients with other primary cancers, or CUPs? How was that certain, by biopsy or post-surgical histology?

Reviewer 3 Report

The topic is of interest and the manuscript is well written. 

It is not fully clear the meaning of brain and torso PET. Were a Brain and torso scan performed additionally to a whole-body PET/CT evaluation in 14222 subjects?

Do the authors think that performing a scan starting from the vertex could lead to similar results? 

Reviewer 4 Report

From an epidemiological and methodological point of view (I'm not a nuclear physician), I have several critical questions for the Authors:

1- it's not clear if Cochrane methodology for systematic review has been followed, can you confirm it? How have you checked the risk of bias? Otherwise, is it a "narrative review"?

2- which search algorithm have you employed? Using MESH or not? If yes, which ones?

3- "We evaluated 14,222 [18F]FDG PET/CT datasets" I'm unable to understand if you analyzed papers or datasets or patients? How have you obtained such an incredible number of datasets? Are you speaking about datasets or patients? maybe the latters

4- "we performed a comprehensive analysis of the international societies’ 
recommendations", either have you analyzed already published reccomandations or datasets? coming from? mixing them? how?

5- "When performing the necessary statistical analyses" this is totally unacceptable, you must write down a full stats methodology section

6- "Shapiro–Wilk test’s results to measure the variables’ distribution" Shapiro test does another job...

7- "H0, Ha, respectively" both these hyphotesis are undefined

8- "3.1. Statistical analyses" this section belongs to M'M, not to results!!!!!

9- "Z – test’s results" to do what? to prove what?

10- "conclusion" I'm still unable to understand why and how a narrative review has been coupled to a so-called experimental section. Suming up official guidelines and personal experience does not add anything to the conclusion, which is quite arbitrary

11- having mixed in an unique pot a review section and an experimental one is totally unacceptable from a Cochrane methodology

12- the suppodsed chenge of management will be applied to an extremely small subcohort of patients. Maybe that a better clinical result could be reached by a more specific radiopharmaceutical compound, rather than 18F-FDG. The conclusions are then unreliable